# Understanding the perspectives and needs of multiple stakeholders: Identifying key elements of a digital health intervention to protect against environmental hazards

Annabelle Workman[1,2], Sharon L. Campbell[1], Grant J. Williamson[3], Chris Lucani[3], David M. J. S. Bowman[3], Nick Cooling[4], Fay H. Johnston[1,5], Penelope J. Jones[1] *

1 Menzies Institute for Medical Research, University of Tasmania, Hobart, Australia, 2 Melbourne Climate Futures, University of Melbourne, Melbourne, Australia, 3 School of Natural Sciences, University of Tasmania, Hobart, Australia, 4 School of Medicine, University of Tasmania, Hobart, Australia, 5 Public Health Services, Department of Health, Hobart, Australia

* penelope.jones@utas.edu.au

**Data Availability Statement:** De-identified data are available on request via the University of Tasmania

## Abstract

AirRater is a free environmental health smartphone app developed and available in Australia that collects individual health data and disseminates environmental hazard information to populations. Following previous evaluations with app users, the aim of this study was to better understand how clinicians, government agency and non-government advocacy group representatives perceive an app designed to reduce the impacts of environmental hazards on individual and public health. Nine government agency and non-government advocacy group representatives, along with 11 clinicians based in Australia participated in a semi-structured interview or focus group to explore perspectives on AirRater. Interview and focus group data were transcribed and analysed using the qualitative data analysis software NVivo. Results indicate that for clinicians, apps like AirRater can add value as an educational, patient self-management and diagnostic tool. For government and peak bodies, apps can add value by addressing environmental health literacy and monitoring and forecasting gaps, as well as supporting advocacy efforts and public health surveillance. We conclude that environmental health smartphone apps can support a range of stakeholders to achieve shared goals and priorities related to individual and public health outcomes. Further research is needed to better understand how apps could be embedded into clinical practice and policy settings.

## Author summary

Health-related smartphone apps are increasingly used to help manage a wide range of range of medical conditions. While many studies focus on evaluating apps from the perspective of individual users, fewer studies explore the viewpoints of other key stakeholders with an interest in health apps, such as clinicians, government agency and non-government advocacy group representatives. In our study, we collected data on how stakeholders

Research Data Portal: https://data.utas.edu.au/metadata/b540bd4e-bd00-42ff-b779-a5b9402ddb98.

**Funding:** This research was funded by the Menzies Philanthropic Fund (to PJ & FJ). The funders had no role in study design, data collection and analysis, decision to publish, or preparation of the manuscript. AirRater is currently operated by AirHealth and is funded by the Department of Health Tasmania, ACT Health, Northern Territory Health, the Australian Department of Home Affairs, and the Menzies Institute for Medical Research, with previous funding by the Northern Territory Environment Protection Authority.

**Competing interests:** The authors have declared that no competing interests exist.

from these groups view the environmental health app 'AirRater'. We used interviews and focus groups to explore their knowledge of, and attitudes toward the app, as well as their insights into barriers for uptake and potential app enhancements. We found that while there are opportunities to enhance the app, key stakeholders consider that AirRater can provide them with valuable information to help support individual and population health outcomes. Our study provides valuable insights into the important role these stakeholders can play in the uptake and reach of an environmental health smartphone app, and the need to consult with a broad range of stakeholders, beyond users, to maximise an app's effectiveness.

## Introduction

A changing climate is increasing many environmental hazards, such as air pollution, pollen, and extreme heat e.g. [1,2,3]. There is an urgent need for interventions that reduce the impact of such hazards on human health, especially in populations considered to be at higher risk of adverse health outcomes from exposure to environmental hazards. These include children, the aged, and those with existing chronic respiratory and cardiovascular conditions, such as asthma, chronic obstructive pulmonary disease, cystic fibrosis and heart disease e.g.[1,4].

Digital technologies, such as smartphone apps, are an important part of the solution to protect individuals from environmental hazards. They can provide accessible environmental hazard information, and tools to understand the relationship between environmental conditions and their health e.g.[5,6]. This in turn can inform individual behaviour change and self-management e.g.[7,8].

In addition, opportunities exist for digital health interventions to also assist non-government stakeholders such as clinicians, public health officers, and health-related organisations, as they seek to achieve shared goals and priorities to maximise individual and community health outcomes [9]. Apps can help clinicians, for example, by providing patient education, and valuable patient data to aid in diagnosis and the monitoring of health outcomes [7]. Apps can also enable government agencies and non-government advocacy groups for population groups with specific health or other needs (known as peak bodies in Australia), to achieve goals with respect to public health outcomes at the population scale by providing public health surveillance and timely, evidence-based messaging, especially to vulnerable populations [10,11].

In other areas of digital health, there is a growing volume of research that explores patient and/or clinician perspectives on the role of apps to support clinical care for specific health conditions e.g. [12,13–21]. This research has had demonstrable benefits: by identifying the needs and preferences of patients and/or clinicians, these studies provide an evidence base for user-centred design of digital health interventions. For example, Ayre et al. utilised in depth interviews with clinicans to identify specific app design needs with respect to diabetes management in culturally and linguistically diverse communities [12], while Slevin et al. identified key barriers and facilitators for the use of digital health technologies to support the management of COPD [19].

Yet, there is a dearth of literature on user perceptions and needs with respect to environmental health apps, particularly with respect to clinicians and broader stakeholders for whom environmental health apps may be beneficial. Such groups include government agencies working to improve environmental health outcomes, and peak bodies advocating for better environmental health policies, programs and services for their members. These additional

stakeholders are rarely considered in discussions of app utility, yet may have diverse reasons for using, valuing and supporting an environmental health app that may present relevant considerations in intervention design. As health-related smartphone apps proliferate, and as health systems around the world become increasingly digitised, it is essential to explore perspectives among clinicians and other key stakeholder groups to maximise the benefits of digital environmental health interventions.

This study seeks to address this knowledge gap with an exploration of clinician, government agency, and peak body perspectives on the AirRater smartphone app, as an exemplar of an environmental health app intervention. AirRater was developed in 2015 by a collaboration including research institutions and government agencies to support Australians affected by air pollution and pollen (for example, those with respiratory or cardiovascular diseases) to protect their health through behaviour change and self-management [10]. The app presents near real-time data captured by government agencies on air quality, pollen count and air temperature at a location. Users can report symptoms, medication use and potential triggers at their current location, and can search an interactive map for air quality, pollen, and temperature data at their own and other locations, as well as monitoring station locations and 'symptom hotspots' based on cumulative user symptom data. A distinctive functionality of AirRater is its aggregation of de-identified symptom data across locations, which can contribute to public health surveillance efforts to inform public health managers.

The app was developed by a multidisciplinary partnership between researchers, clinicians and policymakers and relevant peak body organisations (e.g. Asthma Tasmania), with the aim of incorporating the needs of this diverse range of stakeholders into intervention design. This approach is considered good practice and has been shown to aid community uptake [11]; **Table 1** below outlines the functions that the AirRater system provides for user types from individuals to clinicians, government agencies and peak bodies. App usage accelerated during the 2019–20 Black Summer bushfires on the east coast of Australia, when air quality was dangerously poor. Users of the app reported the benefits of having access to real-time information that was easily accessible and digestible, and highlighted the low levels of air quality literacy throughout the community at the time [6]. Consequently, federal and state inquiries have

**Table 1. Functions and application of AirRater for key stakeholder groups.**

| *Stakeholder* | *Functions and application* |
|---|---|
| Individual end users | • Can monitor hazards, including levels of air pollution, allergens and heat at specific locations via an interactive map and 'traffic light' system to make hazard levels easy to interpret<br>• Can log respiratory symptoms experienced and medication used at a specific time and place<br>• Can view a summary of historical symptom reports over the course of a month, including most commonly reported symptoms, triggers present, use of medication, and time of day most affected by symptoms |
| Clinicians | • Can view (via the AirRater app on patient's phone) the summary of historical symptom reports to advise on treatment and management options |
| Government agency representatives | • Can access de-identified aggregated data from symptom reports to understand the public health burden of environmental hazards, informing public health surveillance and warning efforts |
| Non-government advocacy group (peak body) representatives | • Can use AirRater to support the improvement of literacy on environmental hazards, such as air quality, particularly for people with specific health conditions, such as asthma<br>• Can use de-identified aggregated data from symptom reports to advocate for specific policies and programs to support people with specific health conditions, such as asthma |

delivered recommendations relating to the need for better awareness and education around air quality [22–24]. Further app details and a schematic representation are documented in an earlier publication [10].

AirRater offers an ideal case study to explore clinician, government agency and peak body perspectives on environmental health apps. AirRater has been well characterised through several evaluations that have studied individual user perspectives (as opposed to clinician and broader stakeholder perspectives) through surveys and interviews [5,6]. These evaluations suggest that the app is generally perceived as user-friendly and is effective in supporting individuals to make decisions to protect their health from environmental hazards. Our research has also demonstrated AirRater as a valuable source of epidemiological data [25,26].

In this context, AirRater offers an important opportunity to explore how stakeholders seeking to achieve outcomes at the population level (clinicians, government agency, and peak body representatives) perceive an app designed to reduce the impacts of environmental hazards on individual and public health. This study therefore uses qualitative methods to explore the perspectives of 20 individuals representing these stakeholder groups to contribute to the knowledge base needed to optomise environmental heath app design. Our objectives were to investigate:

- the general attitudes and knowledge of clinicians, government and non-government agency representatives toward AirRater, and

- whether environmental health apps, such as AirRater, can provide these stakeholder groups with an effective clinical tool and/or public health intervention to support positive individual and community health outcomes during changes in air quality and other environmental conditions.

   To meet these objectives, we specifically ask:

- What general attitudes and knowledge do clinical and healthcare professionals have towards health-related smartphone apps, and AirRater in particular? What barriers impact AirRater's effectiveness as a clinical tool and what enhancements could overcome these barriers?

- What general attitudes and knowledge do relevant government agency and peak body representatives have towards health-related smartphone apps, and AirRater in particular? What barriers impact AirRater's capacity as a public health intervention and what modifications could address these barriers?

## Methods and materials

Given our objective to explore unique experiences and perspectives of multiple user groups, qualitative research methods were adopted for this study. A research framework and protocol were developed and informed by the World Health Organization's guidance on monitoring and evaluation digital health interventions [27]. A second qualitative researcher independent of the research team provided feedback. The methodology fulfilled the recommendations towards best practice qualitative health research [28]. The study was approved by the University of Tasmania Health and Human Research Ethics Committee (ID: H0015006). Participants for both groups were recruited from Tasmania, the Australian Capital Territory (ACT) and Port Macquarie in New South Wales (NSW), given AirRater's comparatively high use in these locations as well as recent extensive bushfires and consequential poor air quality in these jurisdictions.

*Clinicians*: Purposive and convenience sampling were used to recruit clinicians. Clinicians were eligible to participate if they were at least 18 years of age and worked in Tasmania, the

ACT or Port Maquarie. There were no exclusion criteria beyond meeting these requirements. All participants were recruited via email and written informed consent was provided prior to participation. One semi-structured interview (see **S1 Table**) was carried out with a clinician based in Tasmania in November 2020. To supplement the interview data, two in-person focus groups were carried out (see **S2 Table**) in Tasmania in March and April 2021. Each focus group comprised five clinicians–nine general practitioners and one practice nurse–who were affiliated with a general practice or allergy clinic. A ten-minute PowerPoint presentation out-lining key features and functionalities of the AirRater app was presented at the beginning of the focus group for the benefit of participants who weren't familiar with the app. The interview length was 34 minutes, and the duration of each focus group was 60 minutes.

*Government agency and peak body representatives*: Purposive and convenience sampling were used to recruit government agency and peak body representatives whose remit included environmental health issues and whose work pertained to Tasmania, the ACT or NSW. Government agency and peak body representatives were eligible to participate if they were at least 18 years of age and worked in, or were undertaking work that pertained to, Tasmania, the ACT or Port Maquarie. There were no exclusion criteria beyond meeting these requirements. Participants were recruited via email and written informed consent was provided prior to participation. All interviews took place via a teleconferencing platform or telephone between September 2020 and April 2021. Semi-structured interviews (see **S3 Table**) were carried out with nine representatives from a relevant government agency or peak body. Recruited participants represented health agencies, environmental protection agencies, meteorological organisations, fire services, local councils, health districts, and peak bodies for respiratory conditions. The average interview length was 31 minutes.

*Data analysis.* All interviews and focus groups were recorded and transcribed verbatim by one researcher (AW) independent of the AirRater research team, and one administrative Air-Rater team member. Once verified for accuracy, all transcripts were uploaded into the qualitative analysis software, NVivo 12 [29]. A blended, deductive-inductive framework approach to coding was employed [30], with the broad themes that comprised the interview schedules for each group initially used as a coding framework to thematically analyse the data. Coding and analysis were undertaken by one researcher (AW) and results were discussed with one other researcher (PJJ) to ensure all relevant themes were described and relevant quotes, including divergent examples, were utilised.

## Results

This section presents results from both clinicians, and government agency and peak body representatives. Results are presented thematically as follows: (i) general attitudes toward health-related smartphone apps, including their potential to address environmental health issues (ii) knowledge of and perspectives on AirRater, including its potential value as a clinical or population health tool, (iii) identified barriers for AirRater as a clinical tool and public health intervention, and (iv) enhancements to address barriers. A summary of results is presented in **Table 2** below, stratified into results relating to (a) clinicians, and (b) government agency and peak body representatives.

### General attitudes towards health-related smartphone apps

**Clinicians.**   Clinicians were familiar with health-related smartphone apps to address a variety of medical conditions, including mental health and diabetes. While personal experiences with apps were varied, clinicians were apprehensive to professionally recommend apps to which they had limited exposure or knowledge:

**Table 2.** *Summary of key results from interviews and focus groups with clinicians, and government agency and peak body representatives, stratified by main themes.*

| | Clinicians | Government agency and peak body representatives |
|---|---|---|
| *Theme 1: General attitudes toward health-related smartphone apps* | • Familiar with numerous apps, most commonly those for managing mental health and diabetes<br>• Reported varied personal app use<br>• In their clinical practice, apprehensive to recommend an app if personal knowledge of the app is limited<br>• Colleagues and reputable sources are key avenues for identifying apps for potential clinical use<br>• Recommended apps to support motivated patients with education, self-management, and health tracking | • Reported varied personal app use<br>• Recognised current and future potential of apps, including as a communication tool to support environmental health outcomes |
| *Theme 2: Knowledge of and perspectives on AirRater* | • Not all clinicians knew of or had recommended AirRater<br>• Many of those that knew AirRater were comfortable to recommended it to patients<br>• Identified value of AirRater for patients as an educational, patient self-management, and diagnostic tool<br>• One clinician indicated that AirRater provided location-specific public health data to inform the treatment of their patients | • Most, but not all representatives had heard of AirRater<br>• Most representatives reported generally positive attitudes toward AirRater, including regarding its free dissemination of information for people with certain health needs<br>• Identified value of AirRater in its capacity to address air quality literacy, monitoring and forecasting knowledge gaps, and advocacy efforts<br>• Identified value of AirRater in its capacity to support public health surveillance and warnings<br>• One representative reported that AirRater was not straightforward to use |
| *Theme 3: Perceived barriers for public uptake of AirRater* | • Similar to their experience with other apps, a lack of interest or commitment from patients to consistently use AirRater and enter all relevant personal data<br>• Patients with poor health and/or general literacy levels are less likely to access and use AirRater in its current design<br>• Patients discouraged by AirRater registration process | • Limited data or knowledge on hazards in specific locations that limit AirRater's utility<br>• Data privacy concerns for end users<br>• Poor air quality literacy and lack of knowledge about AirRater<br>• Poor app literacy, especially in aging populations |
| *Theme 4: Perceived barriers for institution or clinician support of AirRater* | • Time limitations during individual clinical consultations with patients<br>• Limited data and knowledge on local allergens<br>• Patients unable to share data from AirRater with clinicians, impacting the capacity to discuss allergen management<br>• Limited capacity to view historical symptom reporting trends in AirRater | • Institutional hierarchy barriers within health departments, where support is required from executive level staff• Poor air quality literacy and lack of knowledge about AirRater |
| *Theme 5: Enhancements to address barriers* | • Modifications to the AirRater design to facilitate easier data sharing between clinicians and patients<br>• Greater interactivity for patients, such as the capacity to create an asthma plan based on their clinician's recommendations<br>• Creation of AirRater 'Lite' for users with low health and/or general literacy levels<br>• The ability to capture clinical measurements such as peak flow rates in people with asthma to support the monitoring of lung function | • Targeted engagement strategies to:<br>○ embed AirRater into educational settings, such as schools<br>○ connect with respiratory specialists<br>○ connect with hard-to-reach communities<br>• Develop local case studies to showcase and highlight people's experiences and how AirRater has supported end users<br>• Address monitoring network gaps so that location-specific data or knowledge limitations are improved |

". . .there's a few things that I occasionally recommend to people, but I always worry about not knowing enough about them before recommending them." (R4, H)

". . .I don't overly recommend apps to people. . .generally if I'm going to recommend something, I want to have had the time to sit, try it out myself, pretend like I'm the patient, how useful would it be, and a lot of them I just haven't had the time to actually sit down and actually do that." (R5, H)

Clinicians were most comfortable recommending apps they had either used personally, that had been recommended by colleagues or were from sources they considered reputable:

"I would probably go on word of mouth. . .if someone says, 'I've used this app for a patient, it was good', then it's just straight at the top of the list for me. . ." (R5, L)

"...going to a talk or a presentation, people recommending it...that's usually how I've found out about them...often it is something that's been brought up at a talk, get the kind of consensus of the room all going, 'Oh yeah, that's a good one'." (R5, H)

"I look at what's recommended in guidelines...Therapeutic Guidelines is always my first go-to." (R3, H)

"I do use the Digital Health Guide...I can actually email the patient the URL...of the app...I don't think many GPs know about that tool, where you can actually go into the guide, see the review, see it's got five stars, been reviewed by someone who's reasonable, and then cut and paste and send off to the person's email." (R1, H)

Where clinicians did recommend apps, it was primarily to support patients with specific health conditions for education, self-management and health tracking:

"I work in youth health...we have a number of apps to direct young people to some more sort of self-help and self-recognition education...in a professional sense, yes, I certainly have used them...and quite happy to use them." (ID_1)

"...things that I do recommend to people are like *Headspace* and *Smiling Mind*, sort of mindfulness apps..." (R4, H)

## Government agency and peak body representatives

Like clinicians, government agency and peak body representatives were varied in their personal use of smartphone health apps:

"Personally, I minimise my use of apps as much as I possibly can...I prefer to be on a large screen rather than trying to get information from a phone..." (ID_2)

"...I'm not a big app user...mostly my phone is for talking and messaging and taking photos, and I don't do a lot of other apps." (ID_4)

"I'm a subscriber and an enthusiastic subscriber to smartphone apps, and especially health apps...I like them, I use them, I find them useful, and...I would recommend them." (ID_5)

"I think a lot of apps are quite unnecessary but the ones that I use regularly I am very heavily reliant on." (ID_8)

Irrespective of their personal use, government agency and peak body representatives recognised their current and future potential, including as a communication tool:

"I am really interested in terms of how we can evolve them to give people more control of their own health care...I am positive about them; I don't think we've realised the full potential of them yet..." (ID_1)

"I think technology and the use of devices is becoming very wide stream and very accepted and it's certainly improving the way we do business...in terms of communicating but also our ability to gather information..." (ID_3)

"I think they're a great innovation...it's a good way to get real-time information to recipients and all members of the community." (ID_9)

## Knowledge of and perspectives on AirRater

**Clinicians.**   Not all clinicians knew of AirRater, but some indicated that they had recommended it to, or used it with, patients:

"Probably AirRater's the only one I've really recommended with any consistency." (R1, L)

"I use AirRater [as a clinician] quite a lot, but I have very smart patients that come in with graphs of their blood pressure, or graphs of their weight, or their glucose readings. . ." (R4, L)

"I have used AirRater and have recommended that. I haven't had patients *per se* come back and be able to report back to me on how well they've used it, I've used it personally for myself." (R5, H)

Those that had used or recommended AirRater identified several benefits that the app offers patients, including educational, self-management and diagnostic capacities:

". . .the best thing about the app is that you can use it to whatever potential you want. You can go as far as putting medication and stuff in, but you can also use it on a daily basis as, "okay, the pollen count's this, I probably should take some antihistamine"." (R3, L)

". . .I think the app itself has got great opportunity to educate people and help people control symptoms and. . .managing their medications better. . .you can use the app to have people being proactive, proactively managing their own care." (ID_1)

". . .I'm saying, 'Look, you've got birch there and you've got grass there but I'm not quite sure how this is playing, can you have a look at the app and see the pollen counts and then tell me which one, on the days you're bad, which counts are high for which particular pollen'. . .if I'm in doubt as to what to recommend for immunotherapy, it [AirRater] can help." (R4, L)

"I've actually had quite a lot of success with a couple of patients who, where we've done the usual stuff in terms of pollen allergies, and then their management hasn't worked, but their history is so suggestive and their total IgE [antibody] is suggestive of an allergy, and. . .in one case we certainly worked out which pollen it was." (R3, H)

One clinician also indicated that they felt AirRater had a role to play in providing general practitioners with regular location-specific public health data to inform treatment of patients:

"I think we've been sadly let down by the public health's ability to communicate with GPs [general practitioners] what's going on in the health of our community. . .there should be a dashboard that's put up by public health and it's got on there, you know, what the air quality's like. . .are there any other epidemics going on in my region. . .I think AirRater can be part of that data, and that way, all the patients that come in to see me that day, I'm thinking about what is going on, and I'm saying, you know, 'yeah, a lot of people are doing what you're doing, or suffering what you're suffering.'" (R1, H)

## Government agency and peak body representatives

Government agency and peak body representatives were varied in their knowledge of AirRater, from one having no knowledge of the app to others who had been aware of the app since its inception:

"I've been around since AirRater first started, and I've supported it. . .with my work hat on, in terms of getting it off the ground and promoting it and asking questions of it, and seeing what it can do. . ." (ID_1)

"I'm familiar with the concept, I haven't seen it in operation. We've discussed it in various committees and in various forums, and I've seen people present on it. . .as far as the internal workings of the app, no I'm not (familiar). . ." (ID_3)

"The organisation and our stakeholders are all very comfortable and confident about the promotion of the app. . .we promote the app appropriately when the conversation is around air quality." (ID_5)

Those that were aware of AirRater were generally positive about the app, particularly regarding its capacity to support the dissemination of information, although one participant with experience of the app noted some challenges with use:

". . .philosophically I think it's [AirRater] a great idea, I think it's really nice for people to have that easy access to information and I like that it's openly shared, that it's not something that you have to go and pay for. . .it seems to be genuinely focusing on the needs of the person." (ID_4)

"I know that. . .AirRater is useful. . .to a lot of people with. . .breathing difficulties in particular." (ID_7)

"I didn't have a lot of direct experience with AirRater. . .I can really see the benefits of having such a system and, you know, perhaps sort of engaging in a better way than maybe putting a media release out. . .I think it engages people a bit more. . ." (ID_9)

". . .there have been a couple of times. . .when I have used AirRater either to try and find local information or just to see how it works, and my feeling is, to be honest, it's a bit clunky." (ID_8)

Government agency and peak body representatives articulated the value of an app like AirRater as a public health intervention, particularly given its capability to address existing air quality literacy, monitoring and forecasting knowledge gaps, as well as to support advocacy efforts:

". . .our commitment is to supporting an app that can work with our intentions to educate the community around air quality. . .the research capability and the personalisation of advice capability of AirRater is something that is very compelling." (ID_8)

"A lot of people say to us, 'why don't you measure pollen? You should be measuring pollen.' And we say, 'well. . .we're measuring pollutants. We've got a list of pollutants, pollen is not a pollutant as such.' 'Oh, it's important for air quality', and we say, 'well, have a look at AirRater, that's the best we can do.'" (ID_2)

". . .after the summer that we've just had [the 2019–20 Australian Black Summer of extreme bushfires], I think it's really, really clear that national air pollution-type forecasts are something that people want, they need, it would be beneficial to everybody's health to be able to get some more quantitative or spatial information compared to what's currently available." (ID_3)

"...we have our fixed air quality monitoring stations, which we've seen that there are limitations to that fixed network because of the distances between the sensors...potentially something like AirRater would be very useful for filling the gaps..." (ID_3)

"...so we want to know whether, if we're giving advice or warnings or whatever about potential impacts, did those impacts actually happen? ...or were we off base, you know, were we crying wolf...so trying to get that feedback, or data... there's definitely potential to use it [AirRater] in future" (ID_4)

"I guess the other thing is advocacy around air quality generally, you know I think it's a really useful tool to highlight awareness of the fact that we need to do more around air quality..." (ID_1)

Participants also discussed the value of AirRater to aggregate individual user data for epidemiological purposes, and to support public health surveillance and warnings:

"I think the really good thing is that there is now...a really good database of people self-reporting symptoms, and that can be correlated with...what the air quality was doing...that's going to be really, really important into the future..." (ID_2)

"...I think that sort of user interface...engages people as well...knowing that that can then be tracked by the modelling to be able to sort of provide warnings to people, I think that's certainly a great aspect of the app..." (ID_9)

"...if that enables me to go back to departments on the mainland who are saying that they want to set their thresholds at the levels that their experts say are okay, then I can say, 'well actually, this data from actual users tells us that they're having symptoms at a different level.' That's a phenomenal advocacy tool." (ID_8)

## Identified barriers for AirRater as a clinical tool and public health intervention

**Clinicians.** Clinicians identified several existing barriers that impact AirRater's capacity to support patients in a clinical setting. Many clinicians raised broad issues relevant to the use of health apps generally, such as lack of interest or commitment from patients to consistently use an app that has been recommended to them, as well limited time in consultations to discuss apps with patients. Clinicians also raised issues specific to AirRater, for example, regarding the lack of capacity to skin prick test for some of the pollens that AirRater monitors, data entry 'fatigue ', as well as the practical challenges with data sharing via a phone:

"To me most of the issues around using personal devices...is, it depends how much you're ready to commit in terms of putting in data. So even AirRater, are you prepared to put in your medication usage and things like that?" (R1, L)

"I don't think patients put much data into this app...they might do it round, maybe the high pollen season, they might do it for a week if you're lucky, or a couple of weeks, but most patients get sick of it, sick of putting in symptoms, and it bugs them a lot..." (R2, H)

"The problem is we have so little information...we've got five Tasmanian pollens which we've looked at, and where we've done skin pricks, but there are another 'x' number and...we can't test for them..." (R3, H)

"...most people don't have five minutes to talk about an app in a consultation unless you have an hour consult." (R1, H)

"...it's got to be a partnership, where the doctor's given the data and the patient's sitting there with the data together, you're looking at it, on the phone it's really hard, like, you know, we're trying to peer down this little phone, and they're showing us stuff, and it's not quite the same." (R1, H)

Some clinicians identified issues specifically relating to AirRater's current design, with perceptions of education level affecting a patient's ability to engage with the app.

"I have recommended AirRater to practically every patient I see with pollen allergy. I will say that I get very mixed responses, so anything from somebody who has a degree in Science, who has made excellent use of it, to people who don't have a university education who take one look at it and run. And that's one of the problems is the, that personal access." (R3, H)

Other app requirements and features were also identified as potentially impacting AirRater's usability:

"I've found with AirRater that some people might be put off with having to register when you open up the app. . .I don't think it's a huge barrier, but I've just noticed some people have been put out when you first sort of fire up the app to show them and they say, "Oh, I have to put in my personal email address". And it's not immediately clear that you can do it anonymously. . ." (R2, H)

"...when you start. . .to log your symptoms, and you get a symptom report, you can only see from the day you've started, you can't look what the pollen's been doing for the month before you've started logging your symptoms. . .so they [patients] won't see how it's [the pollen levels] been going." (R2, H)

"I think when it comes to putting in triggers, when the patient's self-identifying triggers, so, I notice there's no sort of pollen selection there, as far as I'm aware. You've got cold, dust and exercise, cold but not pollen, or exposure to pollen. . ." (R2, H)

### Government agency and peak body representatives

Government agency and peak body representatives raised organisational, technological, functional and educational barriers as potential impediments to AirRater use that might impact its capacity to be (a) supported and (b) effective as a public health tool:

"...the hierarchical system we work on. . .If it wasn't supported by the wider executive of the environmental health branch. . .it'd be difficult for us at a local level to be. . .specifically promoting something. . .if they were against it. . ." (ID_9)

"...we have been told by people that their information didn't reflect the situation where they were, which I think is really hard, and that's not specific to AirRater, but I think again is, a lot of it will come down to technology advancing. . ." (ID_8)

"...we do have people that are concerned with data privacy and breaches, data privacy is going to be a big thing. . ." (ID_6)

"...a lot of people told us. . .that they just didn't know about AirRater. . .there is no literacy around air quality. . .up until October last year [2020], there was just no understanding

whatsoever. . .I think. . .we still have a long way to go even for people who are very engaged and interested. . .” (ID_8)

“. . .we also have an aged population so they're not, most of them here aren't that savvy with apps I would have to speculate.” (ID_6)

### Enhancements to address barriers

**Clinicians.** Clinicians identified numerous opportunities to enhance AirRater to support its integration into the clinical setting. Some of these suggestions involved modifying the design or functionality of AirRater to facilitate data sharing between patients and clinicians:

“. . .the ability to have the data sent or printed out to the healthcare provider. . .I wonder if that would help to overcome that sort of overwhelming, not. . .placing the impetus on the patient themselves to interpret it, but they gather the data.” (R4, H)

“. . .I think patients would prefer to give permission to share information that they think their healthcare provider will use to interpret, and the GP can interpret what they think's going on, patients like that I think.” (R1, H)

“. . .what I would like more in the app is a bit more interaction with the patient. So. . .to have an asthma plan or a hay fever plan on there, so they've got it there and then they can check in, what we've recommended they do. . .” (R1, H)

“. . .the start of the school year, [I've] constantly got kids coming through the door needing new action plans to be done for school. If you could then link it in with something like this [AirRater], and say to them, 'Oh, this would also be a really great app for your child cause they've got all these allergy-type things', that might be a good way to sort of link it together and just sort of be a trigger for people that every time they do an action plan.” (R5, H)

Other suggestions involved creating an even simpler version of AirRater with limited interactivity for patients with low health literacy levels, as well as the addition of an asthma peak flow meter that can automatically capture lung function data:

“You could call it AirRater Lite and you could just have one screen. . . with symptom, pollens, that's it. . . they could say, 'Thursday, this one was highest, I had lots of symptoms.' That's about the level for a significant number of people. . .and that's about all you need to. . .get that information that we need to try to direct further assessment and treatment. . .” (R3, H)

“. . .if you assume a health literacy of fairly low. . .I think those things need to be explained in quite simple terms. . .with examples. . .maybe even some data demonstration.” (ID_1)

“. . .I think if you don't actually have to actually put data in, you just blow in a peak flow meter and it automatically goes into your data. . .” (R1, L)

### Government agency and peak body representatives

Government agency and peak body representatives also identified several engagement strategies that, if implemented, may support AirRater's capacity as a public health intervention. These included embedding AirRater into key educational environments such as schools,

engaging with medical specialists and hard-to-reach communities, and profiling the app through location-specific case studies:

> "...schools because citizen science is of interest to schools, so it's a good way to engage them...I think there'd be lots of places for this in terms of digital technology, citizen science and health...particularly primary schools but also high schools..." (ID_1)

> "I think that direct engagement with respiratory physicians as a group would be helpful...they would have different perspectives on maybe what it could capture...I think...respiratory physicians because of their...evolving understanding of different types of asthma." (ID_1)

> "...So maybe that's a really important part of a strategy for AirRater is to get into those Facebook groups and...reaching hard-to-reach communities...there is a big need, particularly if there is pollen information given the thunderstorm asthma prevalence among new migrants..." (ID_8)

> "...it would be good if we were able to generate local, highlight local profiles...here's what we've been learning in AirRater about people's experience of the Tasmanian environment in terms of air quality. These are some impacts and...this is how the tool has been benefitting (people)..." (ID_1)

There was also recognition from two participants that addressing monitoring network gaps through both additional resourcing and through explicit messaging within the app would enhance the app's capacity as a public health intervention:

> "...do we have enough detectors so that the tool [AirRater] is sensitive enough to reflect local conditions, and I know that's a bit of a big investment of funds..." (ID_1)

> "...when someone signs up, 'Oh, we notice that you're in an area that doesn't have local air quality monitoring, this is what the information will mean for you, it will be a forecast or a prediction, but your senses are the most important thing, and this is what you need to do.'" (ID_8)

## Discussion

The results of this study show that in an increasingly digitised world, smartphone apps can have a valuable role to play in supporting the needs of key stakeholders beyond individual end users, in the field of environmental health. For clinicians, environmental health apps like AirRater can add value in the clinical setting as an educational, patient self-management, and diagnostic tool. Clinicians reported that the utility of such apps is determined by the level of individual patient commitment to use. They also reported that easy access to data collected is important to maximise an app's clinical utility. Further, clinicians noted that while automated data input makes for a more useful app, additional efforts need to be made to support patients with health and/or other literacy levels recognised as low to initially engage with them. To address this barrier and to increase their overall utility, clinicians recommended embedding apps in other settings, such as schools; and producing simplified versions of the app (eg AirRater 'Lite').

For government agency and peak body representatives, environmental health apps like AirRater can add value as a public health intervention in multiple ways. For example, participants reported the utility of such apps as a powerful communication tool, with the capacity to

address specific gaps in literacy, and to communicate with key subpopulations and hard-to-reach or vulnerable communities. This finding is pertinent given a need to address air quality literacy was established by federal and state inquiries following the 2019–20 Black Summer bushfires in Australia [22–24]. Participants also reported such apps add value by helping to identify and fill existing data gaps–in the case of AirRater, in relation to air quality monitoring and forecasting. Finally, participants reported that such apps can be a powerful advocacy tool by connecting aggregated de-identified individual data with environmental data. Government agency and peak body representatives did report that data privacy remains a concern in relation to apps, which is congruous with findings on clinician perspectives in other studies e.g. [15,21].

Second, our study findings emphasise the importance of addressing existing limitations of some apps through targeted and continued engagement with clinicians, government agency, and peak body representatives whose valuable advice can increase app usability. For example, feedback from clinicians in this study that a 'less is more' app interface, to enable more patients to effectively engage with an app, is consistent with results in other studies e.g.[31,32,33]. Similarly, Karduck et al. (2018) found that approximately a third of practitioners recommended a diabetes self-management app based on a patient's overall literacy level or their health literacy level [34]. This highlights the need to make apps highly accessible and to minimise the burden of data input if they are to be effective in clinical settings and reach diverse population groups. However, app design needs to be carefully considered to ensure that accessibility does not come at the cost of a clinically useful tool. In this context, we suggest that app design involves considerable consultation with both clinicians and potential users from diverse subpopulations and backgrounds.

Third, our study findings align with previous study results regarding limitations around individual patient motivation and capacity to self-manage. Some of the general practitioners involved in our study raised concerns around the motivation required to consistently use an app, which is often necessary to effectively facilitate clinical management. Morrissey et al. (2018) found similar results in their evaluation of general practitioner perspectives on hypertension apps [8]. These results demonstrate the need for incentives for use to be an explicit consideration as part of the app design and consultation process. Gamification is one approach to incentivise engagement in apps, and data to date indicates limitations with the strategy in achieving longer-term user interaction [35]. Beyond the need for further empirical research on gamification theory and design [36], we recommend continued consultation with users and clinicians as an app matures. To address longer-term engagement challenges, introducing incentivisation strategies may be a useful consideration.

Fourth, concerns regarding data privacy and clinical effectiveness raised by participants in this study are reported in findings elsewhere e.g.[18,37], and emphasises the need for integrated and comprehensive guidance to both app developers and users to ensure that apps uphold data privacy expectations and strive to achieve clinical efficacy for users. For AirRater, its development by a consortium of research institutions and government agencies means that strict ethical and data privacy protocols are required and upheld, but these standards are not yet universally mandated for apps developed outside of these settings. Governance of digital health products to monitor and evaluate app data protections and clinical efficacy remains fragmented and inconsistent across countries [38]. An overarching framework that can be adapted to specific jurisdictional legislation is warranted to address governance gaps.

Finally, it is important to note that our study results address a critical gap in the digital health literature, as we have been unable to find evidence that extends smartphone app evaluation to incorporate the perspectives of government agencies or peak body representatives–either within, or beyond, the field of environmental health. As key stakeholders with the

capacity to inform the development of policies impacting population health outcomes, we consider their involvement in app development and evaluation as critical going forward–a principle we consider applicable beyond environmental health to any health domain in which digital health interventions can support the work of such stakeholder groups. Although almost no studies have encompassed the views of such stakeholders to date, beyond AirRater, an evaluation of the health promotion app 'Feed Safe' reported the critical role that interdisciplinary partnerships can play, not only in terms of app development, but also for maximising the reach and uptake of apps in target populations [11].

## Limitations

While our data provide useful and novel perspectives on environmental health apps from diverse stakeholder groups, it is important to note several limitations. First, samples sizes for both groups are small, reflecting difficulty in recruiting as well as time and resource constraints. With clinicians, we overcame challenges recruiting for individual interviews by holding two focus groups; however, for both groups, the sample size precluded us from reaching thematic saturation and limits the generalisability of our findings. However, multiple participant accounts were used in the data analysis rather than relying on a few dominant views. Of note, with respect to government agency representatives, perspectives on perceived opportunities and barriers are likely to be highly context dependent (e.g. health vs air quality agencies, national vs state vs local scale institutions, policy vs operational remits); and our sampling may not have captured all viewpoints. Our results are also grounded in the Australian clinical and institutional setting, therefore application to other contexts must take cultural or institutional differences into account.

## Implications for environmental health app design

Our study presents clear opportunities to enhance or adapt AirRater to maximise its utility to both clinicians and agencies, which may be applicable in other environmental health app contexts. Despite the potential for tension between the needs of users, clinicans and agencies when employing apps for dealing with evironmental hazards, we have found considerable synergy in the attitudes of all these stakeholders towards the current utility and areas of improvement for the AirRater app. Although our findings are most directly applicable to AirRater, we suggest that the general considerations and themes highlighted by our findings may be generalisable with care as a first, initial, set of evidence to inform other environmental health contexts where similar needs and perspectives may be relevant.

First, with respect to clinical settings, the need for simplicity and ease of use is clearly paramount to support integration of AirRater (or other apps) into clinical practice given the patient literacy and consultation time pressures identified. Intelligent, flexible design to minimise literacy barriers, while retaining the variety of functionality needed to support clinician-patient interactions, is therefore important to achieve maximum benefit. Providing clinicians with an easy way to explain what is needed from patients and view their results is critical. This may include developing a dashboard to allow clinicians to easily view environmental conditions and related health outcomes and developing a 'lite' mode within the current app, presenting simple language and key data.

Regarding the needs of agencies, developing systems to allow real time provision of symptom data is important to maximise utility for public health surveillance. To respond to this need the AirRater app has already been updated to utlise the air quality visualisation system, AQVx [39], which draws in population symptom reports from AirRater to a dashboard showing multiple layers of observed and modelled air quality data. Such systems may be of value in

other environmental health contexts a a means of maximising the population-level benefits from environmental health app interventions.

Finally, this study suggests the key design challenge for environmental health apps such as AirRater is to provide the functionality needed to achieve benefits for multiple stakeholders, while maintaining the simplicity of use needed for engagement. More broadly, our study highlights systemic barriers (and opportunities) to the integration of environmental smartphone apps into the increasingly digitised health system. In this context, future research to understand the existing pathways and mechanisms that exist to embed smartphone apps into a variety of health systems would be of high value.

## Supporting information

**S1 Table. Interview schedule for healthcare professionals.**
(DOCX)

**S2 Table. Focus group discussion guide for healthcare professionals.**
(DOCX)

**S3 Table. Interview schedule for agency and peak body representatives.**
(DOCX)

## Acknowledgments

The authors would like to thank Dr Kim Jose for providing feedback on our draft research framework and protocol, and Ms Emerson Easley for her valuable administrative contribution to this research. The authors would also like to thank the 20 clinicians, government agency, and peak body representatives who generously gave their time and perspectives during the study. AirRater collaborators include EPA Tasmania, the Commonwealth Scientific and Industrial Research Organisation, Australian National University, Charles Darwin University, Tasmania Fire Service, Bureau of Meteorology, Asthma Australia and the University of Melbourne. Since 2023 AirRater has been operated by AirHealth, a joint venture between the University of Melbourne and the University of Tasmania.

## Author Contributions

**Conceptualization:** Annabelle Workman, Fay H. Johnston, Penelope J. Jones.

**Data curation:** Annabelle Workman.

**Formal analysis:** Annabelle Workman.

**Funding acquisition:** Fay H. Johnston, Penelope J. Jones.

**Investigation:** Annabelle Workman.

**Methodology:** Annabelle Workman, Nick Cooling, Fay H. Johnston, Penelope J. Jones.

**Project administration:** Annabelle Workman, Nick Cooling.

**Software:** Grant J. Williamson, Chris Lucani.

**Supervision:** Fay H. Johnston.

**Writing – original draft:** Annabelle Workman.

**Writing – review & editing:** Annabelle Workman, Sharon L. Campbell, Grant J. Williamson, Chris Lucani, David M. J. S. Bowman, Nick Cooling, Fay H. Johnston, Penelope J. Jones.

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
