## [Decision Letter · Decision Letter 0]

6 Jul 2023

PDIG-D-23-00174

Meeting the needs of multiple stakeholders: Identifying key elements of a digital health intervention to protect against environmental hazards

PLOS Digital Health

Dear Dr. Jones,

Thank you for submitting your manuscript to PLOS Digital Health. After careful consideration, we feel that it has merit but does not fully meet PLOS Digital Health's publication criteria as it currently stands. Therefore, we invite you to submit a revised version of the manuscript that addresses the points raised during the review process.

Please submit your revised manuscript within 60 days Sep 04 2023 11:59PM. If you will need more time than this to complete your revisions, please reply to this message or contact the journal office at digitalhealth@plos.org. Please include the following items when submitting your revised manuscript:

We look forward to receiving your revised manuscript.

Kind regards,

Haleh Ayatollahi

Section Editor

PLOS Digital Health

Journal Requirements:

1. We have noticed that you have uploaded Supporting Information files, but you have not included a list of legends. Please add a full list of legends for your Supporting Information files after the references list.

2. In the online submission form, you indicated that "Requests for access to de-identified data presented in this study can be made to the corresponding author". All PLOS journals now require all data underlying the findings described in their manuscript to be freely available to other researchers, either 1. In a public repository, 2. Within the manuscript itself, or 3. Uploaded as supplementary information.

Additional Editor Comments (if provided):

The manuscript was interesting and well-written. Please address the following issues in your revision.

1- Please present the methods section after the introduction and before the result section.

2- Please provide a table of themes, subthemes, categories, etc. to show the summary of results. In fact, Table 1 needs to be revised to be more informative.

3- Please explain how the researchers examined the trustworthiness of the qualitative results.

4- Please follow the journal instructions for citation within the text.

Reviewers' comments:

Reviewer's Responses to Questions

**Comments to the Author**

1. Does this manuscript meet PLOS Digital Health’s publication criteria? Is the manuscript technically sound, and do the data support the conclusions? The manuscript must describe methodologically and ethically rigorous research with conclusions that are appropriately drawn based on the data presented.

Reviewer #1: Partly

Reviewer #2: Yes

Reviewer #3: Yes

Reviewer #4: Yes

2. Has the statistical analysis been performed appropriately and rigorously?

Reviewer #1: N/A

Reviewer #2: Yes

Reviewer #3: N/A

Reviewer #4: N/A

3. Have the authors made all data underlying the findings in their manuscript fully available (please refer to the Data Availability Statement at the start of the manuscript PDF file)?

Reviewer #1: No

Reviewer #2: Yes

Reviewer #3: Yes

Reviewer #4: Yes

4. Is the manuscript presented in an intelligible fashion and written in standard English?

Reviewer #1: Yes

Reviewer #2: Yes

Reviewer #3: Yes

Reviewer #4: Yes

5. Review Comments to the Author

Reviewer #1: Can you explain the term Peak Body for non Australian audiences? i.e., non-governmental organisations, representative body etc 

In the introduction, can you be more specific about the background research and the benefits demonstrated through the research into this space.

Make it clearer what the current evidence shows and the gap in existing information that provides the justification for your study.

Make it clearer what the aims and objectives of your study are.

Add more information regarding how the participants were selected. Was any inclusion / exclusion criteria of participants considered?

What methodology was used to agree the coding system? Was this informed by any previous research?

Can you present the main coding themes that formed the basis of your results?

Review and improve the approach used for citing - (e.g. 1, 4). Perhaps the editor could comment better on this.

Suggest moving methods before results so that the reader can better follow how the research was undertaken prior to reading the results.

Describe / define what is meant by ‘vulnerable populations’

Can you better align the questions you present with the headings used in the results?

For consideration – the authors may want to consider more granularity in presenting the results, separating the results into the aspects of the app stakeholders felt were useful and to whom, how this can deliver patient value - and what aspects were seen as negatives or barriers? e.g. benefits might be: early warning of poor air quality, improving health literacy, improved clinical symptom management.

Did clinician skills come up as an issue and if so how can this be addressed?

Add to the discussion….

Any conflicting view from those identified in previous research? If not, explore possible reasons for any conflicting views?

Reviewer #2: This paper appears to be a reasonably well written qualitative evaluation of the AirRater app, which appears to provide some interesting data and can be useful to the groups interviewed in the paper. While I don't have a lot of specific comments regarding the paper, I have some larger general ones which I think the authors should address with revisions. The sample size of 20, even though it is fragmented across a few groups, is also better than other papers I have seen lately for qualitative research.

General comment 1: What kind of clinicians/patients (ie. what illness) are intended to use the AirRater app? When I was going through the paper the exact audience for the app was unclear to me.

General comment 2: Why no images of app in the draft, or explanation of its dashboards? (perhaps they were in supporting documentation?)

General comment 3: Its good you have a table summarizing your findings under results, I just wonder if there is a way to pack more information into something like that considering that you have a lot of interview text to discuss.

General comment 4: You swap "AirRater" for "environmental health smartphone apps" in a few places, such as in the abstract. Are you talking about apps "like AirRater" or is AirRater such a good example that it well-represents all "environmental health smartphone apps"? If so, justify, because the paper is somewhat unfocused in terms of addressing your research questions; you tend to jump around from an evaluation of "AirRater" to generalisations about "environmental apps" and the paper could probably be tightened up overall if you spend some time on this. Another example would be at the start of your discussion.

Introduction/Abstract - Line 79 (for instance) - Please define "peak bodies" for international readers, or change the term to something more generic. This appears to be an Australian term.

Introduction/Abstract - Line 105 - Further app details and a schematic representation are documented elsewhere (10). - Can you please elaborate on what this means?

Discussion - Line 408 - I would change "confirm" to "shows".

Discussion - Line 463 - A critical gap in the literature with respect to apps or air quality apps?

Discussion - Line 478 - "rather" - spelling

Discussion - Line 490 - "utility" - spelling

Reviewer #3: Dear Dr Haleh Ayatollahi, 

Meeting the needs of multiple stakeholders: Identifying key elements of a digital health intervention to protect against environmental hazards. 

The proposed manuscript is the summative report of the findings related to a digital health intervention to protect citizens against environmental hazards. In my opinion, the paper was well written, and the following should be addressed in writing due to specific reasons:

• Upfront, the authors should state the rationale for selecting the stakeholders. To this end, state what was the selection criteria of the stakeholders. Why was it essential to choose these stakeholders for his app? It needs to be clarified why these stakeholders were selected. If you address this as a multiple-stakeholder study, why is there more theoretical framing on clinicians? It is unclear why the one stakeholder in terms of literature and findings was given more voice and scaffolding in the paper, even why you separated the results as a clinician, government agency, and peak body representative. Is the app of greater priority to one stakeholder than the other? 

• This description of the needs identified should be indicated earlier in the paper. This paper needs to clarify the rationale for the interface of a multi-stakeholder analysis. 

• Indicate the critical elements for each stakeholder and include what need this fulfils for each stakeholder. It needs to be evident from the results how this app identifies key aspects of critical intervention. Instead, the paper is clear concerning knowledge and attitude and the barriers when stakeholders engage with the app. Consider revising the abstract and title. 

• After reading the questions, there appears to be a mismatch between the title, results reported and the questions. 

See: 

We specifically ask:

• What knowledge and attitudes do clinical and healthcare professionals have towards health-related smartphone apps in general, and AirRater in particular? What barriers hinder AirRater from providing maximum utility in a clinical setting, and what enhancements could overcome these barriers?

• What knowledge and attitudes do relevant government agencies and peak body representatives have towards health-related smartphone apps in general and AirRater in particular? What barriers impact AirRater’s capacity as a public health intervention, and what modifications could address these barriers?

• To the authors, if you are asking questions about the knowledge and attitudes, how does this relate to meeting the needs of these stakeholders? I suggest revising the title and abstract, considering the congruence not present between the research question and results in the paper. 

I propose that the authors address the minor changes to revise the manuscript based on the above-referenced feedback. Therefore, minor revision is required to consider for publication.

Reviewer #4: This is a small, limited but still useful investigation of multi-stakeholder responses to a smartphone app that combines air monitoring with personal health data. It both offers behavioral guidance for users and collects data for public health surveillance and research. Developing tools for these functions is an important direction for environmental health and public health. Identifying barriers to adoption by clinicians is particularly important since they are gatekeepers and trusted communicators. The interest by government officials is encouraging.

6. PLOS authors have the option to publish the peer review history of their article (what does this mean?). If published, this will include your full peer review and any attached files.

**Do you want your identity to be public for this peer review?** For information about this choice, including consent withdrawal, please see our Privacy Policy.

Reviewer #1: Yes: Shoshana Bloom

Reviewer #2: No

Reviewer #3: No

Reviewer #4: No

---

## [Decision Letter · Decision Letter 1]

7 Jan 2024

Understanding the perspectives and needs of multiple stakeholders: Identifying key elements of a digital health intervention to protect against environmental hazards.

PDIG-D-23-00174R1

Dear Dr Jones,

We are pleased to inform you that your manuscript 'Understanding the perspectives and needs of multiple stakeholders: Identifying key elements of a digital health intervention to protect against environmental hazards.' has been provisionally accepted for publication in PLOS Digital Health.

Best regards,

Haleh Ayatollahi

Section Editor

PLOS Digital Health

Reviewer Comments (if any, and for reference):

Reviewer's Responses to Questions

**Comments to the Author**

1. If the authors have adequately addressed your comments raised in a previous round of review and you feel that this manuscript is now acceptable for publication, you may indicate that here to bypass the “Comments to the Author” section, enter your conflict of interest statement in the “Confidential to Editor” section, and submit your "Accept" recommendation.

Reviewer #1: All comments have been addressed

Reviewer #2: All comments have been addressed

Reviewer #3: All comments have been addressed

2. Does this manuscript meet PLOS Digital Health’s publication criteria? Is the manuscript technically sound, and do the data support the conclusions? The manuscript must describe methodologically and ethically rigorous research with conclusions that are appropriately drawn based on the data presented.

Reviewer #1: Yes

Reviewer #2: Yes

Reviewer #3: Yes

3. Has the statistical analysis been performed appropriately and rigorously?

Reviewer #1: N/A

Reviewer #2: Yes

Reviewer #3: N/A

4. Have the authors made all data underlying the findings in their manuscript fully available (please refer to the Data Availability Statement at the start of the manuscript PDF file)?

Reviewer #1: Yes

Reviewer #2: Yes

Reviewer #3: Yes

5. Is the manuscript presented in an intelligible fashion and written in standard English?

Reviewer #1: Yes

Reviewer #2: Yes

Reviewer #3: Yes

6. Review Comments to the Author

Reviewer #1: Many thanks for redrafting the manuscript, I can see suggested improvements have been actioned and have improved the publication considerably.

Reviewer #2: The edits improved the paper overall and addressed my initial comments. I still could not find an actual image of the app anywhere, I looked at the supplementary link but it was just a .docx file - perhaps I missed it.

Reviewer #3: Thank you for attending to the comments with such detail and thoughtfulness. The authors have taken great lengths to respond to the feedback. It is evident from the changes and the justification provided that they strengthened the paper's writing and the argument's legibility.

7. PLOS authors have the option to publish the peer review history of their article (what does this mean?). If published, this will include your full peer review and any attached files.

**Do you want your identity to be public for this peer review?** For information about this choice, including consent withdrawal, please see our Privacy Policy.

Reviewer #1: **Yes: **Shoshana Bloom

Reviewer #2: No

Reviewer #3: No
